# OpenReview forum: "TRACEALIGN - Tracing the Drift: Attributing Alignment Failures to Training-Time Belief Sources in LLMs"
_ICLR.cc/2026/Conference — ICLR 2026 Conference Withdrawn Submission_

### Official Review · Reviewer_NQHf · 2025-10-28

**Soundness:** 2
**Presentation:** 1
**Contribution:** 2
**Rating:** 2
**Confidence:** 4

**Summary:**

This paper proposes a framework called TRACEALIGN, designed to trace and mitigate alignment drift in large language models. The authors argue that model misalignment does not primarily stem from prompt attacks during reasoning, but rather from “belief conflicts” lingering from the training process. The framework includes a corpus tracing module (TRACEINDEX), a semantic conflict metric (BCI), and intervention methods covering reasoning, training, and decoding stages. The paper also introduces a benchmark, ADB, covering sensitive domains such as hacking, self-harm, and hate speech. Experimental results show that the framework can significantly reduce drift while maintaining performance. Overall, the paper aims to shift alignment research from purely behavioral concerns to the level of model beliefs.

**Strengths:**

- The paper presents a relatively novel perspective, focusing on the alignment of internal model beliefs rather than external metrics like refusal rates or toxicity scores.
- The framework design is relatively complete. TRACEINDEX and BCI complement each other: the former identifies the source of generated content, and the latter measures semantic conflicts, forming a clear causal chain.
- The multi-layered defense strategy proposed in the paper covers reasoning, training, and decoding, offering practical reference value.

**Weaknesses:**

1. The introduction is poorly structured; the core methods are not sufficiently described and the paper jumps directly to results, making it somewhat unclear.
2. The dataset, a key contribution of the paper, is described too briefly. A more detailed account of data sources, selection criteria, and annotation methods is needed to make the contribution convincing.
3. Figures and tables are poorly laid out, with some placed far from the corresponding text. Except for Figure 2, other figures are not cited in the main text, making it unclear what they correspond to. Figure 3’s text is too small to read. Figure captions should describe the framework rather than presenting result data. Overall, the layout is cluttered, harming readability.
4. The methodology section lacks logical flow, with fragmented paragraphs that read like notes. It is recommended to first provide an overview of the framework before describing each module, giving readers a global understanding.
5. The authors mention the impact of TRACEINDEX’s frequency threshold on results but provide no experimental validation or parameter analysis, lacking evidence.
6. The theoretical analysis, while interesting, is not rigorous. The so-called “upper bound on drift probability” lacks clear definitions or assumptions. Some symbols are used inconsistently across sections, e.g., τ and ℓ_q are reused with different meanings, making it appear more like a heuristic conjecture than a formal conclusion.
7. Currently, only RLHF and DPO are used as baselines, without including recent alignment and safety strategies such as Constitutional AI [1], RAFT [2], or model-editing-based methods like ROME [3] or MEMIT [4]. These methods represent different alignment approaches and are necessary for a meaningful comparison to validate TRACEALIGN’s advantages. Moreover, the evaluation metrics are limited, reporting only drift rate and perplexity (PPL), lacking more informative measures such as factual consistency, alignment score, refusal accuracy, or safety violation rate. Introducing such metrics and comparing across multiple baselines would substantially enhance the credibility and reproducibility of the experimental results.

> [1] Bai, Y. et al. Constitutional AI: Harmlessness from AI Feedback. arXiv:2212.08073, 2022. \
[2] Dong, Q. et al. RAFT: Reward-Augmented Fine-Tuning for Generative Foundation Models. ICLR 2024. \
[3] Meng, K. et al. ROME: Locating and Editing Factual Associations in GPT. NeurIPS 2022. \
[4] Meng, K. et al. MEMIT: Mass Editing Memory in a Transformer. ICLR 2023.

**Questions:**

see weaknesses.

---

### Official Review · Reviewer_FFjz · 2025-10-28

**Soundness:** 2
**Presentation:** 2
**Contribution:** 2
**Rating:** 4
**Confidence:** 3

**Summary:**

This paper argues that most of the harmful completions in the presence of the adversarial texts can be traced back to memorization in the pre-training. Building upon this argument the paper introduces a method based on suffix matching to retrieve less frequented samples from the pre-training corpus which might have led to the harmful completion. The less frequented nature of the samples are measured by a negative log frequency based score (BCI). Based on this score the paper proposes an inference and training time mitigation in the face of adversarial prompt. In the inference time the paper proposed to mediate refusal based on the BCI score of the completion. In the training time defense the work proposes to penalize DPO samples based on the BCI score on the assumption that those DPO samples may reinforce the memorization that occurred in pre-training.

**Strengths:**

The idea of tracing vulnerabilities to memorization in the dataset is interesting. This as a finding serves as a contribution for this paper.

The paper also contributes by the proposal of an adversarial dataset for stress testing alignment drift in large language models.

This finding indeed can serve as a potential future direction of motivation for unlearning certain memorization sources given the identification of a vulnerability in post deployment. But the paper delves into using this memorization metric itself as a measure in fine tuning or inference to mediate refusal.  This brings my concerns into the practicality as mentioned in the weakness section.

**Weaknesses:**

1. Given the he inference time defense is doing a suffix matching through the entire pre-training corpus

     *  How feasible is it as a practical defense given that the pre-training corpus should be scanned at all inference times
     * Beyond harmful refusal wouldn’t this also prevent memorization on essential tasks. Also how does this defense prevent over-refusals. In the ORBench style prompt how does this method score?
2. Similarly in the preference fine tuning, how much does this strategy penalize the useful samples (for instance in mathematical tasks etc would’t this penalize useful answers/ expressions which were seen in pre-training dataset. Eg Quantum equation etc). Beyond perplexity can you give performance penalties in benign tasks to justify the utility preservation.

3. The paper lacks study into utility preservation in non toxic tasks and over refusal which questions the practicality of the contribution as a viable defense/ fine tuning.

4.  Formating concerns: The margin formatting of the paper should be fixed in rebuttal.

**Questions:**

See weaknesses

---

### Official Review · Reviewer_Yrsj · 2025-10-31

**Soundness:** 3
**Presentation:** 2
**Contribution:** 3
**Rating:** 4
**Confidence:** 4

**Summary:**

The paper introduces TRACEALIGN, a provenance-centered framework to diagnose and mitigate alignment drift in LLMs by tracing unsafe generations back to memorized spans in pretraining data; it builds TRACEINDEX, a suffix-array span index (with approximate matching) and defines a Belief Conflict Index (BCI) that scores risk using token frequency–based cross-entropy, then deploys three complementary defenses—TRACESHIELD (inference-time refusal on high-BCI spans), a Contrastive Belief Deconfliction (CBD) loss added to preference fine-tuning, and Prov-Decode (decoding-time veto with provenance), all evaluated on a curated Alignment Drift Benchmark (ADB) covering high-risk domains; combined, these components reportedly cut drift by up to ~85% while preserving fluency and refusal quality, reframing alignment as “what models remember and believe” rather than only surface behavior.

**Strengths:**

1. Clear motivation: reframes “alignment drift” as training-data belief conflict rather than purely behavioral metrics.

2. Concrete toolkit: unified pipeline (indexing → BCI scoring → decoding/fine-tuning defenses) with an at-a-glance figure and component ablations.

3. Actionable defenses: TRACESHIELD and Prov-Decode are operationally simple and produce explainable refusals tied to retrieved spans.

**Weaknesses:**

1. Violations of template policy: The submitted PDF lacks the required footer page numbers on each page. This indicates the authors modified the official template (apparently to reclaim space), which violates the Call for Papers formatting rules. I recommend the PC treat this as a format noncompliance; even setting aside technical merits, the paper should not be accepted in its current form.
2. Novelty issues: Core attribution relies on suffix-array exact-span matching (OLMoTrace lineage). The advance over existing provenance tools feels incremental (mainly a new risk score + three wrappers), not a fundamentally new attribution mechanism.
3. Limited application scenarios: Methods require access to (unsafe) pretraining corpora; for most closed or proprietary models this is infeasible, reducing external impact.

**Questions:**

1. Template compliance: Why were footer page numbers removed? Please restore the official template and confirm total pages remain within limits.
2. Closed-model settings: What’s the recommended path when training data isn’t accessible? Could a weaker but deployable variant work with public surrogates?

---

### Note · Authors · 2026-01-09

I have read and agree with the venue's withdrawal policy on behalf of myself and my co-authors.